# Mutation in Hemagglutinin Antigenic Sites in Influenza A pH1N1 Viruses from 2015–2019 in the United States Mountain West, Europe, and the Northern Hemisphere

**DOI:** 10.3390/genes13050909

**Published:** 2022-05-19

**Authors:** Craig H. Decker, Naomi Rapier-Sharman, Brett E. Pickett

**Affiliations:** Department of Microbiology and Molecular Biology, Brigham Young University, Provo, UT 84602, USA; craigdecker1@gmail.com (C.H.D.); naomi.rapier.sharman@gmail.com (N.R.-S.)

**Keywords:** influenza virus, H1N1, hemagglutinin, HA, comparative genomics, virology, bioinformatics, phylogenetic tree, selection pressure

## Abstract

H1N1 influenza A virus is a respiratory pathogen that undergoes antigenic shift and antigenic drift to improve viral fitness. Tracking the evolutionary trends of H1N1 aids with the current detection and the future response to new viral strains as they emerge. Here, we characterize antigenic drift events observed in the hemagglutinin (HA) sequence of the pandemic H1N1 lineage from 2015–2019. We observed the substitutions S200P, K147N, and P154S, together with other mutations in structural, functional, and/or epitope regions in 2015–2019 HA protein sequences from the Mountain West region of the United States, the larger United States, Europe, and other Northern Hemisphere countries. We reconstructed multiple phylogenetic trees to track the relationships and spread of these mutations and tested for evidence of selection pressure on HA. We found that the prevalence of amino acid substitutions at positions 147, 154, 159, 200, and 233 significantly changed throughout the studied geographical regions between 2015 and 2019. We also found evidence of coevolution among a subset of these amino acid substitutions. The results from this study could be relevant for future epidemiological tracking and vaccine prediction efforts. Similar analyses in the future could identify additional sequence changes that could affect the pathogenicity and/or infectivity of this virus in its human host.

## 1. Introduction

The H1N1 subtype of Influenza A virus that was responsible for the 1918–1919 influenza pandemic killed between 50–100 million individuals [1]. More recently, the 2009 triple-reassortant swine H1N1 influenza A (pH1N1) virus was also associated with a pandemic with a much lower mortality rate [2]. Specifically, this recent pandemic was responsible for approximately 60 million infections and an estimated 12,469 deaths [3]. Although nearly a century elapsed between these two H1N1 pandemics, ongoing research efforts enabled a rapid response when the pH1N1 virus emerged. These past pandemics highlight the continued importance of sequence-based surveillance efforts to facilitate mutation tracking and evolutionary predictions that enable the development of more robust and effective vaccines against future influenza A isolates [4,5].

Influenza A virus (IAV) possesses several features that enable it to be a perennial threat to human health. The high error rate of the IAV RNA-dependent RNA polymerase [6], the reassortment of the eight genomic segments [7,8], the perpetual ability to infect humans and other hosts [9,10], and evasion of the host immune response [11]. IAV consists of eight genomic segments that traditionally code for at least ten proteins, with the potential for producing at least seven additional proteins [12]. The Hemagglutinin (HA) segment is 1698 bases in length, codes for a protein that is 565 amino acids long and is extremely immunogenic. The HA protein is known to interact with the sialic acid on the surface of human cells, facilitating viral entry into host cells [13]. Consequently, the properties and overall structure of the HA sialic acid-binding region is highly conserved among IAVs to facilitate entry into the host cell [14,15]. In contrast, epitope regions in the HA protein are constantly under positive selection pressure from the host adaptive immune system [16], which is detected by observing higher-than-expected mutation rates at antigenic sites [17].

It is imperative to continue performing sequencing and surveillance in all regions of the world in an effort to reduce the risk that one or more advantageous mutations spread and become predominant, as documented with the pH1N1 [18,19,20]. Since the emergence of the 2009 pH1N1 triple-reassortant virus, a range of mutation and case-rate tracking studies have been conducted in many countries worldwide, including Australia [21,22], China [23], India [24,25], New Zealand [21], Singapore [21], the United Kingdom [26], the United States [27], and Zambia [28].

Through the combination of surveillance efforts with high-throughput sequencing technologies, the influenza research community has meticulously tracked viral protein changes across subtypes and lineages. Early efforts identified at least three epitope regions in the HA protein [29,30]. In 2010, Maurer-Stroh et al. noted an E391K mutation in the pH1N1 HA sequence that destabilized the protein by altering salt-bridge interactions [31]. Sakabe et al. demonstrated that the D127E, K142N, and D222G HA mutations in pH1N1 were necessary to adapt to a mouse host [32]. Ginting et al. showed that the combination of the neuraminidase H274Y mutation and the T82K, K141E, and R189K HA mutations resulted in oseltamivir resistance and increased virulence, thus augmenting the danger of the emergent 2009 H1N1 pandemic strain [33]. An analysis of H1N1 mutations concluded that HA substitutions improved protein stability that was lost after the prior emergence of seven mutations at functional regions [18].

The aim of this comparative genomics study was to identify sequence substitutions in the HA protein that emerged between 2015–2019. Specifically, those that play a key role in the evolutionary trajectory of IAV across diverse geographic regions. While prior studies have evaluated and compared H1N1 mutational trends during this time period [34], they have not compared viruses collected in the United States Mountain West to other regions and have included both covariance and selection-pressure analyses.

## 2. Materials and Methods

### 2.1. Sequence Datasets

A bioinformatics workflow was designed to facilitate the analysis of HA sequence data from 2015–2019 (Figure 1). HA sequence data for the selected geographical areas were retrieved from the Influenza Research Database (www.fludb.org, accessed on 1 March 2020; IRD) [35]. The search criteria consisted of HA coding sequences from H1N1 viruses collected between 2015 and 2019 from avian, human, and swine hosts. Additional search parameters were applied to construct four sets of sequences based on the geographical location of sample collection (Figure 2), including: (A) Mountain West (MW) dataset consisting of HA sequences from Arizona, Colorado, Idaho, Montana, Nevada, New Mexico, Utah, and Wyoming; (B) USA dataset with HA sequences from Alabama, Arizona, Colorado, Georgia, Idaho, Louisiana, Mississippi, Montana, Nevada, New Mexico, Texas, Utah, Vermont, Washington state, and Wyoming; (C) European dataset with HA sequences from Belgium, Czech Republic, Denmark, France, Germany, Italy, Russia, Spain, and the United Kingdom; as well as (D) a Northern Hemisphere HA sequence set consisting of the three collections described above as well as representative sequences from Japan and Canada. Sequences that contained unreasonably large numbers of insertions/deletions or other errors were manually excluded from our datasets and subsequent analysis. Maps highlighting the regions included in each dataset were made using MapChart [36].

### 2.2. Sequence Alignment and Variant Identification

Each of these four nucleotide-sequence sets was first aligned with MAFFT version 7.471 [37] using default parameters, followed by visualization and manual inspection with JalView version 2.11.1.0 [38]. Computational translation of the nucleotide sequences was performed to generate amino acid sequences that were subsequently realigned with MAFFT. The amino acid alignments were then used as input to the metadata-driven comparative analysis tool for sequences (meta-CATS) [39]. Briefly, this algorithm performs a chi-square statistical analysis on aligned sequences to identify positions that display a statistically significant skew in the distribution of residues and the associated metadata (e.g., geographical or temporal point of isolation). Amino acid positions were mapped to the HA protein using the A/California/04/2009 strain as the reference sequence. Positions identified by meta-CATS as significant (*p*-value < 0.05) were included in subsequent analyses and indicated a potential association between one or more sequence variations and a given metadata attribute, such as time of virus isolation.

The Shannon entropy for each alignment was calculated using the integrated IRD tool to quantify the prevalence of amino acid substitutions across each flu season in each dataset [40]. This provided a quantitative value for overall sequence diversity. The sequence feature variant type component of the IRD was queried to find annotated functions associated with each position containing an amino acid substitution [41]. The meta-CATS and entropy data were used to track the progress of the variants in structural, functional, and/or epitope regions through geographic areas.

### 2.3. Phylogenetic Tree Reconstruction

The Mountain West dataset was first evaluated with the 3seq algorithm [42]. This analysis determines whether recombinant sequences were present since such sites can bias downstream phylogenetic reconstructions and selection-pressure analyses [43,44]. Default settings were used to analyze 520 nonidentical HA CDS sequences, where both inferred segments had a minimum length of at least 100 nucleotides. Randomized Axelerated Maximum Likelihood-Next Generation (RAxML-NG) version 0.9.0 was run in a high-performance computing environment to generate bootstrapped maximum-likelihood phylogenetic trees for each aligned set of nucleotide sequences using the GTR-G model [45]. The Robinson–Foulds (RF) distances were used to measure the similarity of symmetric Maximum-Likelihood (ML) trees generated by RAxML-NG from the multiple sequence alignments. Relative RF distances were calculated [46] with smaller absolute and relative distances representing more similarity between the generated tree and the best-scoring ML tree (Appendix A). Bootstrapped trees were then generated for each geographically distinct set of sequences and the RF distance was calculated.

### 2.4. Selection-Pressure Analysis

Prealigned sequences used for selection pressure were directly retrieved from the IRD website. Each set of aligned sequences was used to create a maximum-likelihood phylogenetic tree using PhyML software [47] on the IRD website with the HKY evolutionary model and all other default parameters. These IRD trees were used as seeds for the HyPhy analysis. HyPhy version 2.5.1 (MP) for Linux was run in a high-performance computing environment to detect positive or negative selection pressure on the aligned HA nucleotide sequences [48]. This analysis was performed using the mixed-effects model of evolution (MEME) [49], fixed-effects likelihood (FEL) [50], and single-likelihood ancestor counting (SLAC) algorithms. [50,51]

For each aligned sequence set, the selection-pressure analysis used the following procedure: phylogenetic tree files (in Newick format) and HA nucleic acid CDS aligned sequence files (in fasta format) were downloaded from IRD and the stop codons were manually trimmed from each alignment. Sequences that decreased the accuracy of the alignment were identified and manually excluded from this analysis. The trimmed sets of sequences were then run through HyPhy. The MEME algorithm was used to identify specific sites that were undergoing selection pressure [49]. The FEL method was used to test each site in the alignment for an overall evolutionary rate [50]. The SLAC algorithm generated a theoretical common ancestor coding sequence and then used it to calculate the synonymous and nonsynonymous substitution rates [50]. The output of all selection pressure algorithms was manually combined and compared to minimize false positives due to algorithm bias(es).

### 2.5. Bayesian Evolutionary Analysis

Bayesian Evolutionary Analysis by Sampling Trees (BEAST) v2.6.3 was used for Bayesian phylogenetic tree reconstruction. This Markov chain Monte Carlo (MCMC) resampling algorithm uses a Bayesian method to generate a posterior probability distribution that estimates the correctness of the tree based on the provided data. Aligned nucleotide files from each dataset (MW, USA, Euro, Northern Hemisphere) were converted from fasta to Nexus format to meet input requirements. BEAUti v.2.6.4 was used to generate the XML-based input file and a relaxed clock model was employed [52]. BEAST was run in a high-performance computing environment [53].

Effective sample size (ESS) is a measure of the quality of the BEAST analysis. In order to reach an ESS of more than 200—the commonly accepted BEAST quality indicator—the datasets were run with a chain-sampling frequency of 1000 for varying chain lengths [54]. The wide variation present in these large datasets required chain lengths of 100 to 450 million generations (Appendix A) to achieve acceptable ESS values, with values from 100–200 considered acceptable in some scenarios and values greater than 200 considered ideal. In cases where multiple tree files were generated for the same sequence set, they were combined with BEAST’s postprocessing tool Logcombiner v2.6.3 and viewed with Tracer v1.7.2 [55]. Two of the resulting trees (MW and Euro) were strict consensus trees with 10% of states removed, while the other two (USA and Global) were majority-rule consensus trees with 50–60% of states removed due to the large number of sequences in the datasets [56]. An ESS of ≳200 for three of the datasets was calculated by Tracer with the NH dataset having an ESS of 112 (Appendix A). Finally, the trees were combined to create a consensus tree using TreeAnnotator v2.6.3 and viewed with FigTree v1.4.4.

### 2.6. Coevolutionary Sequence Analysis

The Mutual Information Server to Infer Coevolution (MISTIC) was used to calculate coevolution, or covariance, between pairs of aligned amino acid positions in the HA protein [57]. This method uses a mutual information (MI) algorithm to identify residues that coevolve or co-vary with others in the aligned sequences. The output was summarized in tabular form and as a circos plot displaying the MI coevolution network.

## 3. Results

### 3.1. Basic Analytical Design

For each dataset, we followed the same bioinformatics analytical workflow (Figure 1). First, we gathered all available H1N1 sequence samples that met our search criteria in the IRD from the geographical region being studied (MW, USA, Euro, or NH; Figure 2, Table 1). We then reconstructed phylogenetic trees to determine the evolutionary relationships between H1N1 HAs over time. To better understand the sequence variations that play an important role in separating the phylogenetic clades, we next performed a meta-CATS analysis on these aligned sequences to identify amino acid substitutions that significantly differed between 2015 and 2019. This method incorporates a chi-square statistical approach, with the sequences from each region divided into two groups (pre-summer 2017 and post-summer 2017). This approach enabled us to identify which amino acid positions displayed significant “skew” in the distribution of residues before and after the summer of 2017. We specifically chose the summer in the Northern Hemisphere as the division, since it corresponds to a natural break between flu seasons. The meta-CATS method reports amino acid substitutions that significantly associate with the year of collection, and not when the amino acid substitutions occurred. We consequently calculated Shannon entropy values, as implemented in the IRD, for each alignment to determine when the amino acid substitutions occurred within our 2015–2019 temporal window. This method calculates the frequencies of amino acids present at each position across an alignment of flu sequences. For example, HA position 200 had 57% proline and 43% serine during the period between November 2016 to March 2017. Using the Shannon entropy values, we created tables to facilitate tracking of substitutions occurring at each position, when they occurred temporally, and their frequencies.

### 3.2. 2015–2019. Mountain West H1N1 HA Sequences Fall into Two Distinct Clades with Specific Substitutions

We began our analysis of the Mountain West (MW) dataset by using the 3seq method to confirm that no recombinant sequences were present in this dataset. We then reconstructing maximum likelihood (Appendix A) and Bayesian (Appendix A) phylogenetic trees for this dataset, with the Bayesian tree having an ESS value of 291.6. These nucleotide tree reconstructions confirmed two primary clades, the first consisting of strains isolated in 2015–2016 and the second consisting of strains collected in 2018–2019. Given this phylogenetic signal, we next used meta-CATS to identify sequence positions that most contributed to the phylogenetic topology. This analysis identified 56 statistically significant amino acid positions that differentiated strains collected between the 2015–2016 Northern Hemisphere flu seasons (i.e., pre-summer 2017), and the 2017–2018 Northern Hemisphere flu seasons (i.e., post-summer 2017) (Appendix A). Sorting these positions by their *p*-values and subsequent manual review led us to prioritize three amino acid substitutions that are all located in functional and epitope regions of the HA protein: K147N (HA1 130), P154S (HA1 137), and S200P (HA1 183). We also identified positions 159 (HA1 142) and 233 (HA1 216), which are both in functional regions, as having strong *p*-values with some potential statistical skew introduced by low numbers of certain amino acid residues at these positions. We also identified eight additional positions that are located primarily in known immunogenic regions, including 62 (HA1 45), 177 (HA1 160), 179 (HA1 162), 190 (HA1 173), 252 (HA1 235), 277 (HA1 260), 299 (HA1 282), and 313 (HA1 297). We then used the HyPhy software to determine whether these codons were subjected to selection pressure. We observed positive selection pressure at position 200 as well as the eight positions in immune epitope regions. In contrast, we observed no pressures at positions 147, 154, 159, or 233 (Appendix A).

### 3.3. USA, Euro, Northern Hemisphere Sampling Results Overview

After completing this process, we wanted to confirm our findings while minimizing the contribution of sampling bias from sequences collected in the US Mountain West region. We consequently decided to evaluate progressively larger datasets from more broad geographical regions (e.g., United States, Europe, and the Northern Hemisphere). Due to the large number of 2015–2019 H1N1 samples available in the United States, it was intractable to analyze all samples. To ameliorate this difficulty, in addition to the sequences from the MW dataset, we included representative samples from a variety of states across the United States, Europe, and the Northern Hemisphere.

Specifically, we wanted to determine whether the K147N, P154S, and/or S200P substitutions, and to a lesser extent the substitutions at positions 159 and 233, could be detected before and after the summer of 2017 in each of the larger sets of sequences. We decided to focus on the substitutions at positions 147, 154, and 200 since they are located within functional regions that were not solely characterized as immune epitopes. We consequently excluded the eight positions in known epitopes, as well as positions 159 and 233, since we observed a low number of counts for a subset of amino acid residues at these latter two positions, which adversely skewed statistics and did not justify a more in-depth temporal analysis.

We therefore created new datasets (Figure 2, Table 1) by sampling US states outside of the Mountain West region, nine countries from Europe, and additional Northern Hemisphere countries. We then reran meta-CATS on these datasets to identify any significant substitutions that were detected at these positions. We observed that many of the same 56 sites that contained significant changes in the MW set maintained their statistical significance in the three other datasets (Appendix A). Specifically, we observed 70, 48, and 161 sites that contained significant changes between pre-summer 2017 and post-summer 2017 in the United States, Europe, and NH datasets, respectively. We also observed no selective pressures at positions 147, 154, 159, or 233. Position 200 had detectable pressure by the FEL and SLAC algorithms in the MW dataset, and by the SLAC algorithm in the USA dataset (Appendix A). The maximum-likelihood and Bayesian phylogenetic trees for the USA (Appendix A), Euro (Appendix A), and NH (Appendix A) datasets continued to show well-separated clades in the 2015–2016 and in the 2018–2019 temporal periods. We observed that the ESS values for the more divergent datasets were lower than expected, which is not surprising given the high mutation rate for influenza viruses. Specifically, the ESS values for the USA, European, and Northern Hemisphere trees were 197.8, 607.9, and 112, respectively.

### 3.4. Selection Pressure

Our selection-pressure analysis across the various sequence sets identified 27 HA residues that underwent either positive or negative selection pressure in at least one regional dataset (Table 2; Appendix A). We observed that position 200 underwent detectable selection pressure only in the United States and Mountain West datasets. We also detected statistically significant evidence selection occurring at the eight immune epitope positions, including positions 62, 177, 179, 190, 252, 277, 299, and 313. In contrast, we found that positions 147, 154, 159, and 233 were not detected by this analysis.

### 3.5. Temporal Prevalence of Mutation across Geographical Areas

We next wanted to determine a more precise evolutionary timeline of the non-epitope positions, or those with skewed *p*-values, by calculating the frequency of amino acid residues that were at positions 147, 154, and 200 (Table 3, Table 4 and Table 5). To accomplish this, we generated Shannon Entropy values for each amino acid position across the multiple sequence sets. We observed that the S200P mutation was relatively uncommon in Europe before the 2017–2018 flu season, with a frequency of 28.13% between 2015 and 2017 (Table 5). In contrast, the frequency of this substitution increased during the 2018–2019 flu season, reaching a frequency of 93.51% in Europe towards the end of the 2019 flu season. These quantitative results provide additional insight into the increased prevalence of the S200P substitution across each dataset in our study to eventually become the dominant allele. Subsequent visualization of these substitutions in a three-dimensional HA structure confirms their respective location and relevant biological functions (Appendix A).

### 3.6. Additional Amino Acid Positions with Significant Temporal Changes

One mutation of particular interest is the nonsynonymous S200P substitution. The meta-CATS analysis for the NH dataset calculated a *p*-value of 9.308 × 10^−237^ for the S200P amino acid substitution, which is located within the sialic acid-binding domain of HA.

The K147N substitution was detected in only 13.8% of the Mountain West sequences by the 2018–2019 influenza season, but still achieved a *p*-value of 8.323 × 10^−13^. In contrast, over the summer of 2018, we calculated that 6% of the H1N1 sequences in the Northern Hemisphere dataset had a lysine-to-asparagine substitution at this position (Table 3), with a *p*-value of 3.404 × 10^−21^.

The substitution at position 154 underwent a partial change in our datasets from proline to serine during 2015–2019 (Table 4), with a *p*-value of 3.404 × 10^−21^ in the NH dataset. We observed that the partial P154S substitution first gained traction in the European dataset during the 2016–2017 (60%) and 2017–2018 (13%) flu seasons, then emerged in the Mountain West (25%) and the USA (23%) datasets in the 2018–2019 flu season.

Positions 159 and 233 had significant *p*-values in the Northern Hemisphere comparison (0.0016 and 6.28 × 10^−40^, respectively). However, as mentioned above, these positions displayed biased *p*-values due to the skew in the observed amino acid residues at these two positions. Due to the expected change in epitope regions over time, we did not calculate the temporal change of the eight epitope regions that we identified above.

### 3.7. Coevolution

We next wanted to determine whether any compensatory mutations facilitated the three substitutions described above. We consequently performed a coevolution analysis using the public MISTIC server. We observed that residues 147, 154, and 200 had large mutual information (MI) values with multiple other residues (Table 6; Appendix A), which indicates that they may have coevolved with other positions as a cluster. We generated a circos plot to visualize the large number of coevolving residues (Appendix A). Additionally, we observed several distinct large MI values for residues coevolving with positions 147 and 154. Since the largest MI values have the highest rates of coevolution, these changes can be assumed to be non-independent. Although we did not observe strong quantitative evidence of direct coevolution between most pairwise combinations of our five selected non-epitope residues, we did observe that the forty-first best-scoring coevolution result was between positions 147 and 233. We did observe strong coevolution values for seven of the eight epitope positions, including 62, 177, 190, 252, 277, 299, and 313.

### 3.8. Amino Acid Positions Having Substantial Coevolution, Selection Pressure, and/or Temporal Changes

We then examined the remaining 53 of 56 amino acid positions in HA that had FDR-corrected *p*-values < 0.05 (Appendix A) as well as functions annotated from the literature. We filtered this original list to include those that were most likely to drive a sustained change over time. Specifically, we focused on positions located in either functional or protein structural regions rather than epitope regions—primarily since nearly all positions in the HA protein have already been characterized as immune epitopes. This analysis identified two positions, 159 and 233, which had significant meta-CATS *p*-values. Residue 233 displayed strong evidence of coevolution (Table 5), while position 159 showed some evidence of coevolution (Appendix A).

We also observed other positions in functional and/or structural regions that had been identified by at least two of the algorithms. Positions that significant Bonferroni-adjusted meta-CATS *p*-values as well as good coevolution values include 147, 154, 202, and 233. Those positions with significant adjusted meta-CATS *p*-values and significant HyPhy *p*-values include 3, 6, 155, and 203. Lastly, positions with significant adjusted meta-CATS *p*-values, acceptable coevolution values, and significant HyPhy results include 179, 190, and 200 (Appendix A).

Lastly, we detected amino acid positions in known human immune epitope regions that had significant adjusted meta-CATS *p*-values within our temporal period, significant HyPhy *p*-values for selection pressure, and measurable coevolution values in at least one comparison. Specifically, these positions include 62, 177, 179, 190, 252, 277, 299, and 313.

## 4. Discussion

The goal of this study was to identify sequence substitutions that contributed to the observed separation of pH1N1 HA protein sequences from the Mountain West region of the United States and other regions around the world into two distinct phylogenetic clades. We then confirmed that these two clades are present in multiple sets of sequences from diverse geographic regions across the Northern Hemisphere. Subsequent sequence analyses for selection pressure and coevolution identified amino acid positions 200, 154, and 147 as strongly contributing to this divergence, with these positions playing roles in protein function, structure, and/or immune epitopes. We identified positions 159 and 233 as having a measurable contribution, and other positions with a smaller contribution to sequence divergence in IAV in the Northern Hemisphere between 2015–2019.

To our knowledge, applying a computational selection-pressure analysis to pH1N1 datasets from various geographical regions during this temporal period is novel. The advantage of using the three chosen selection-pressure algorithms is their ability to detect different facets of a complex system; with each method having unique capabilities to identify relevant codons undergoing selection. Similarly, the application of a coevolution method further supported a more in-depth analysis of the residues at positions 147, 154, 200, and 233. The lack of statistical significance for the S200P substitution in the European and Northern Hemisphere datasets could be due to various factors, including the increased prevalence of codons that produce this substitution in the European and Northern Hemisphere populations.

Our statistical analysis of amino acid sequences identified HA position 200 as relevant to the clade separation, which has been characterized previously [34]. Specifically, residue 200 has been shown to be located within the highly conserved sialic acid-binding site [58,59,60]. Position 200 is also located within a B-cell epitope in mice [61], rabbits [61], humans [62], as well as within a T-cell epitope [63]. Whether or not position 200 tests positive or negative as a T-cell epitope likely depends on the exact sequence surrounding position 200 in the strain in question and which HLA allele is present in the host [64,65,66,67].

Typically, HA functional pockets undergo relatively low levels of mutation [68], making the emergent S200P substitution somewhat surprising. The substitution of proline for any other amino acid has been shown to potentially impact the folding, secondary structure, and/or stereochemistry of a protein due to its ring structure [69]. Interestingly, Al Khatib et al. (2018) discovered that S200P increases binding avidity between HA and sialic acid [34]. The potentially large effect of the S200P substitution on the structure of the highly conserved binding pocket likely justifies additional experimental investigation.

Previous studies have also noted the presence of the S200P mutation in pH1N1 HA proteins [70]. The S200P substitution first appeared in Uganda during late 2010/early 2011 [71] and in the United States in 2010 [34]. Based on this evidence of S200P emerging in the early 2010s, it is possible that the surge of S200P that occurred during 2015–2019 was a second wave, accompanied by compensatory mutations elsewhere in the protein that may have increased its fitness [72]. Researchers observed a similar surge in China from 2017–2019 [73].

HA position 154 is located within the highly conserved sialic acid binding site, which is located near position 200 in three-dimensional space. This position partially underwent a change from proline to serine during the summer of 2017 [74]. This previous observation at least partially supports the possibility that P154S coevolved with the S200P mutation and may compensate to optimize function after the emergence of the former change.

Our observation that large frequencies of S200P substitutions are often accompanied by P154S substitutions is interesting since the latter is located in a T-cell epitope [75,76] a B-cell epitope [62] the sialic acid binding site [77]. The P154S mutation has been shown to allow efficient escape from antibodies targeting the Ca2 antigenic region around residue 154 [60]. The P154S HA mutation, which appeared to decrease viral replication of oseltamivir-resistant H1N1 strains in ferrets [78], emerged as early as 2010 in Germany [79].

The 10% K147N shift during the summer of 2018 represents a mutation that may justify continued investigation and surveillance. We found this substitution interesting because it occurs within the functional Lysine fence [80]. This sequence plays a role in stabilizing the virus–host membrane interaction during entry into a new host cell. To our knowledge, it is unknown whether this substitution affects the function of the lysine fence. Additionally, HA position 147 is located within known B-cell [62] and T-cell [63,76,81,82,83,84,85] epitopes. This antigenic shift has been associated with immune escape [60]. Previous passaging experiments of the A/California/07/09 isolate in the presence of a novel neuraminidase inhibitor gave rise to a subpopulation of viruses that coded for the K147N substitution, suggesting a potential selective advantage for HA proteins containing the K147N substitution [86].

Our identification of positions 159 (HA1 142) and 233 (HA1 216) is of particular interest. Specifically, the former is found in the conserved Lysine fence region [80,87], which has been shown to be unique to pH1N1 strains [87]. This infers that it would not be subjected to selection pressure. Its lack of substantial coevolution suggests that this position evolved independently of others and may be at least a partial driver of the antigenic drift that we observed before and after the summer of 2017. Position 233 is located in the receptor-binding site [88]. The lack of selection pressure and the detected coevolution at this site indicates that changes at this site may have been accompanied by other substitutions across that HA protein with an overall preference for sequence conservation. It is likely that the five sites identified by this study substantially contributed to the continued virulence of Influenza virus in 2017. Although positions 159 and 233 had highly significant *p*-values, the statistics were likely at least somewhat skewed due to the presence of low numbers of residues (<3) in at least one of the groups of sequences that were compared.

Positions 3, 6, 154, 155, 202, 203, 179, and 190 had attributes of being important to the divergence of IAV; however, we are unable to determine whether they are likely to be drivers of the antigenic drift or simply co-occurring passenger substitutions. Additional experiments in the wet lab are needed to better quantify whether they affect viral fitness and/or pathogenesis.

Amino acid positions 62, 177, 179, 190, 252, 277, 299, and 313 are located within known human B-cell and/or T-cell epitope regions. Interestingly, position 179 lies within the well-characterized S_a_ epitope region first identified by Caton et al. [29,89]. Although it is not surprising to identify regions undergoing selection and coevolution as epitope regions, such substitutions could hint at which epitope regions changed to facilitate continued infection of various hosts in our selected temporal period.

Phylogenetic tree reconstructions have been generated previously for IAV strains collected during our target temporal period [58,90]. Our observation that Influenza A viruses undergo recombination rarely has been reported previously [42,91]. The phylogenetic analyses that we performed showed a distinct separation of clades between strains isolated in the Northern Hemisphere before and after 2017. The fact that our trees obtained good ESS values across multiple datasets supports that influenza A viruses evolve under a molecular clock [92,93,94]. We suspect that the large shift in the prevalence of strains that contained these three substitutions occurred between the 2016–2017 and 2017–2018 Northern Hemisphere seasons. A subset of the strain metadata only specified the year of isolation, which prevents us from providing a more precise time when this separation of clades became readily apparent. We expect that future analyses with additional sequence data and in-depth metadata will improve our understanding of when this phenomenon occurred.

## 5. Conclusions

Our phylogenetic results show that a substantial antigenic drift event occurred before and after 2017 in the pH1N1 HA protein. Comparative sequence analysis showed significant changes occurring at HA positions 147, 154, 159, 200, and 233 after the summer of 2017 in the Northern Hemisphere. A subset of these positions was found to undergo selection pressure and to coevolve as a cluster, which likely improved overall fitness. We expect these findings could be relevant to ongoing evolutionary studies, as well as future vaccine-strain prediction efforts.

## Figures and Tables

**Figure 1 genes-13-00909-f001:**
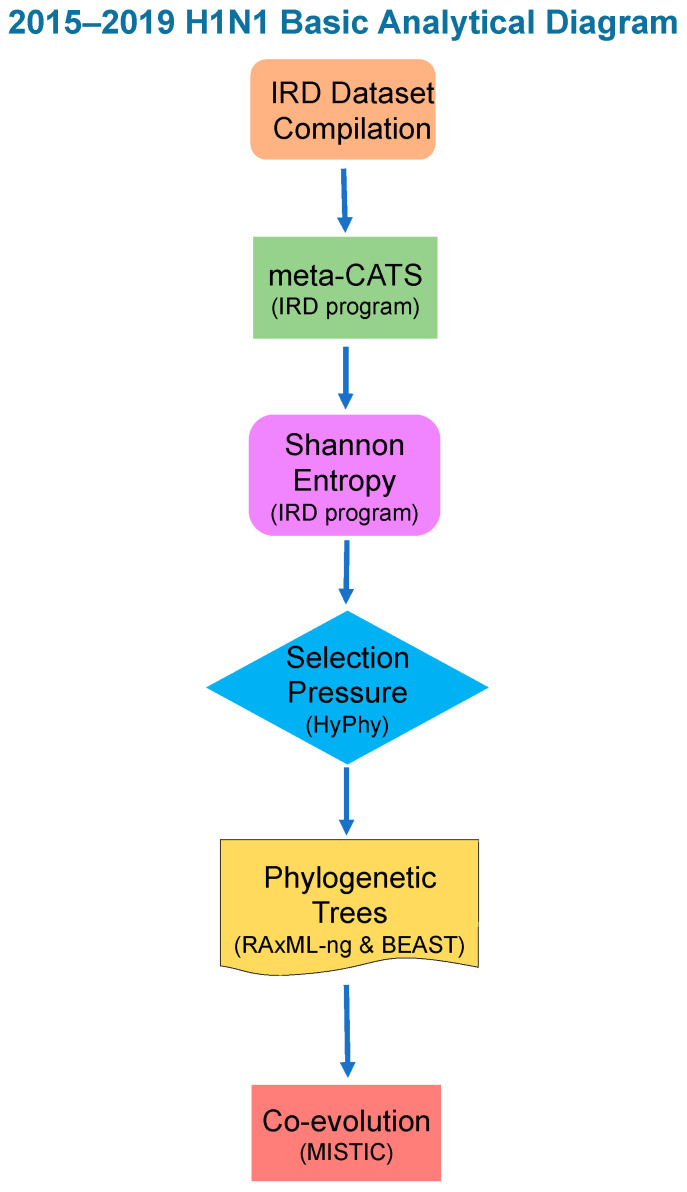
A visual representation of the analytical workflow implemented in the current study.

**Figure 2 genes-13-00909-f002:**
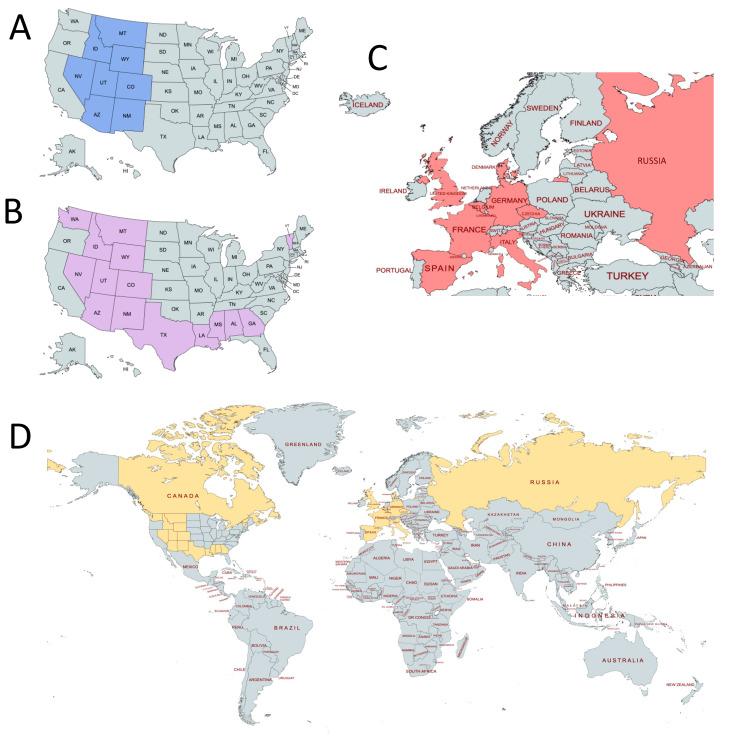
Geographical regions represented by each dataset. (**A**) Mountain West dataset; (**B**) USA dataset; (**C**) European dataset; (**D**) Northern Hemisphere dataset.

**Table 1 genes-13-00909-t001:** Total number of samples in each dataset.

Mountain West	USA	Europe	Northern Hemisphere
933	1924	309	2389

**Table 2 genes-13-00909-t002:** Significant results from each HyPhy selection pressure algorithm by codon position, with the MW, US, EU, and NH datasets represented in blue, lavender, red, and yellow (respectively).

MW MEME	MW FEL	MW SLAC	US MEME	US FEL	US SLAC	EU MEME	EU FEL	EU SLAC	NH MEME	NH FEL	NH SLAC
3			3	3	3				3		
	6	6	6	6	6						
			11	11	11	11	11	11	11	11	11
36											
						57					
							60				
65			65								
			86								
90			90								
			137						137		
145											
	146										
	156	156		156	156						
											158
180	180										
200	200	200			200						
				203	203		203	203	203	203	203
						273					
			278	278	278						
289											
			472								
505											
						513			513		
						524	524				
545						545	545		545	545	545
						550	550			550	
						566					

**Table 3 genes-13-00909-t003:** Evolutionary timeline of hemagglutinin K147N variant by flu season and dataset, with the MW, US, EU, and NH datasets represented in blue, lavender, red, and yellow (respectively).

	MW		US		EU		NH	
Year	%K	%N	%K	%N	%K	%N	%K	%N
Summer 2015	100% (5/5)	0% (0/5)	100% (11/11)	0% (0/11)	100% (25/25)	0% (0/25)	100% (37/37)	0% (0/37)
Winter 2015–2016	100% (297/297)	0% (0/297)	100% (568/569)	0% (0/569)	100% (64/64)	0% (0/64)	100% (665/666)	0% (1/666)
Summer 2016	100% (34/34)	0% (0/34)	100% (59/59)	0% (0/59)	100% (4/4)	0% (0/4)	100% (66/66)	0% (0/66)
Winter 2016–2017	100% (69/69)	0% (0/69)	100% (143/143)	0% (0/143)	90% (9/10)	10% (1/10)	99% (153/154)	1% (1/154)
Summer 2017	100% (12/12)	0% (0/12)	96% (25/26)	0% (0/26)	NA *	NA *	96% (25/26)	4% (1/26)
Winter 2017–2018	100% (122/122)	0% (0/122)	100% (316/317)	0% (0/317)	% (53/53)	0% (0/53)	100% (469/470)	0% (1/470)
Summer 2018	97% (28/29)	3% (1/29)	93% (42/45)	0% (0/45)	100% (1/1)	0% (0/1)	94% (47/50)	6% 3/50
Winter 2018–2019	91% (292/322)	8% (26/322)	90% (595/658)	9% (56/658)	95% (116/122)	5% (6/122)	91% (722/791)	8% 62/791
Summer 2019	44% (19/43)	56% (24/43)	55% (53/96)	44% (42/96)	100% (30/30)	0% (0/30)	67% (86/129)	33% 42/129

K: lysine; N: asparagine; winter season includes October through March; Summer season includes April through September. MW: Mountain West region: US: USA region; EU: Europe region; NH: Northern Hemisphere region. * NA values were recorded for absence of sequences during specific time periods.

**Table 4 genes-13-00909-t004:** Evolutionary timeline of hemagglutinin P154S variant by flu season and dataset, with the MW, US, EU, and NH datasets represented in blue, lavender, red, and yellow (respectively).

	MW		US		EU		NH	
Year	%P	%S	%P	%S	%P	%S	%P	%S
Summer 2015	100% (5/5)	0% (0/5)	100% (11/11)	0% (0/11)	60% (15/25)	32% (8/25)	70% (26/37)	24% (9/37)
Winter 2015–2016	99% (293/297)	1% (4/297)	99% (563/569)	1% (6/569)	84% (54/64)	9% (6/64)	98% (650/666)	2% (12/666)
Summer 2016	91% (31/34)	9% (3/34)	95% (56/59)	5% (3/59)	50% (2/4)	25% (1/4)	92% (61/66)	6% (4/66)
Winter 2016–2017	100% (69/69)	0% (0/69)	99% (142/143)	1% (1/143)	40% (4/10)	60% (6/10)	95% (147/154)	5% (7/154)
Summer 2017	100% (12/12)	0% (0/12)	100% (26/26)	0% (0/26)	NA *	NA *	100% (26/26)	0% (0/26)
Winter 2017–2018	98% (120/122)	2% (2/122)	98% (311/317)	2% (6/317)	87% (46/53)	13% (7/53)	95% (447/470)	5% (22/470)
Summer 2018	97% (28/29)	3% (1/29)	93% (42/45)	7% (3/45)	0% (0/1)	100% (1/1)	92% (46/50)	8% (4/50)
Winter 2018–2019	75% (240/322)	25% (80/322)	76% (503/658)	23% (153/658)	94% (115/122)	6% (7/122)	80% (629/791)	20% (160/791)
Summer 2019	93% (40/43)	7% (3/43)	91% (87/96)	9% (9/96)	100% (30/30)	0% (0/30)	93% (120/129)	7% (9/129)

P: proline; S: serine; winter season includes October through March; Summer season includes April through September. MW: Mountain West region: US: USA region; EU: Europe region; NH: Northern Hemisphere region. * NA values were recorded for absence of sequences during specific time periods.

**Table 5 genes-13-00909-t005:** Evolutionary timeline of hemagglutinin S200P variant by flu season and dataset, with the MW, US, EU, and NH datasets represented in blue, lavender, red, and yellow (respectively).

	MW		US		EU		NH	
Year	%S	%P	%S	%P	%S	%P	%S	%P
Summer 2015	100% (5/5)	0% (0/5)	100% (11/11)	0% (0/11)	64% (16/25)	36% (9/25)	73% (27/37)	27% (10/37)
Winter 2015–2016	99% (295/297)	1% (2/297)	100% (567/569)	0% (2/569)	84% (54/64)	16% (10/64)	98% (654/666)	2% (12/666)
Summer 2016	91% (31/34)	9% (3/34)	95% (56/59)	5% (3/59)	50% (2/4)	50% (2/4)	92% (61/66)	8% (5/66)
Winter 2016–2017	43% (30/69)	57% (39/69)	55% (79/143)	45% (64/143)	20% (2/10)	80% (8/10)	53% (82/154)	47% (72/154)
Summer 2017	83% (10/12)	17% (2/12)	81% (21/26)	19% (5/26)	NA *	NA *	81% (21/26)	19% (5/26)
Winter 2017–2018	25% (31/122)	75% (91/122)	19% (61/317)	81% (256/317)	75% (40/53)	25% (13/53)	33% (153/470)	67% (316/470)
Summer 2018	14% (4/29)	86% (25/29)	16% (7/45)	84% (38/45)	0% (0/1)	100% (1/1)	18% (9/50)	82% (41/50)
Winter 2018–2019	4% (13/322)	96% (309/322)	3% (22/658)	97% (636/658)	7% (9/122)	93% (113/122)	4% (31/791)	96% (760/791)
Summer 2019	5% (2/43)	95% (41/43)	3% (3/96)	97% (93/96)	3% (1/30)	97% (29/30)	3% (4/129)	97% (125/129)

P: proline; S: serine; winter season includes October through March; Summer season includes April through September. MW: Mountain West region: US: USA region; EU: Europe region; NH: Northern Hemisphere region. * NA values were recorded for absence of sequences during specific time periods.

**Table 6 genes-13-00909-t006:** Mutual information coevolution for the 20 highest-scoring pairs of residues *.

1° Residue Position	1° Residue	2° Residue Position	2° Residue	Mutual Information Value **
181	S	312	I	1175.141846
91	S	181	S	1163.426392
91	S	312	I	1159.654785
**200**	**S**	**312**	**I**	**714.461975**
**91**	**S**	**200**	**S**	**710.796936**
**181**	**S**	**200**	**S**	**661.189087**
**146**	**N**	**202**	**T**	**606.855347**
**146**	**N**	**277**	**N**	**527.068542**
**62**	**R**	**315**	**I**	**462.919434**
**299**	**P**	**315**	**I**	**450.615295**
**202**	**T**	**277**	**N**	**435.101715**
**62**	**R**	**299**	**P**	**423.069916**
**154**	**P**	**468**	**N**	**382.397064**
**147**	**K**	**313**	**H**	**374.515015**
**154**	**P**	**190**	**V**	**370.027618**
**252**	**E**	**537**	**V**	**356.437103**
**147**	**K**	**177**	**K**	**323.149689**
**177**	**K**	**313**	**H**	**316.047058**
**177**	**K**	**233**	**T**	**316.031616**
421	I	523	E	307.183044

***** Bold text emphasizes rows containing at least one position from the current work (e.g., 62, 147, 154, 177, 190, 200, 202, 233, 252, 277, 299, and 313). ** Larger MI values signify higher levels of coevolution.

## Data Availability

Publicly available data were analyzed in this study. The consensus HA sequences used in this study can be found at www.fludb.org (accessed 1 March 2020).

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
