# Peer review of "Mutation in Hemagglutinin Antigenic Sites in Influenza A pH1N1 Viruses from 2015–2019 in the United States Mountain West, Europe, and the Northern Hemisphere"

_genes, 2022, doi:10.3390/genes13050909_

Round 1

Reviewer 1 Report

The supplementary metherials are some confuse. The phylogenetic trees includs humans, birds and swine HA of influenza viruses.

Therefore phylogenetic trees looks like unusial and bulky. I even not imagine - how it may be publish.
Perhaps the authors may change these trees - to choose less number of viruses and use only human viruses. It will be better reflected the main aim of this article

Author Response

Reviewer Comment 1: The supplementary metherials are some confuse.

Response: We thank the reviewer for their helpful input about the supplementary materials. We have updated the numbering of the supplementary materials and have added tables S1 and S2 that were referenced in the text.

Reviewer Comment 2: The phylogenetic trees includs humans, birds and swine HA of influenza viruses. Therefore phylogenetic trees looks like unusial and bulky. I even not imagine - how it may be publish.

Response: We agree that these phylogenetic tree graphics can be difficult to read when viewed at normal size. One of the primary reasons that we decided to include these tree visualizations as pdf-formatted supplemental files was to enable the reader to zoom in and agree with our findings.

We purposefully included humans, birds, and swine in our analysis in an attempt to capture the sequence variation in the HA protein among various hosts that are known to be responsible for the inter-species spread of influenza virus.

While the phylogenetic trees created in this study do contain many sequences, this density of sequence information provides branch point values to compare specific sequences. One of the main goals of this work was to spotlight the grouping observed in the trees based on virus time of isolation. Grouping of specific years demonstrates generally how the virus is evolving temporally within these geographical regions. We believe that the patterns visualized in these larger trees helps to reinforce our findings that a large-scale antigenic drift event occurred between the spring and fall of 2017.

Reviewer Comment 3: Perhaps the authors may change these trees - to choose less number of viruses and use only human viruses. It will be better reflected the main aim of this article

Response: We thank the reviewer for making us aware of this potential issue. As mentioned earlier, we included multiple host species to adequately represent the variation in the population of HA sequences among various hosts. This allowed us to more accurately track the emergence of changes in the HA protein over time. In an effort to reduce the number of HA sequences that were included in this work, we also reconstructed a phylogenetic tree that includes only sequence data from the Mountain West region of the United States. This graphic enables the reader to visualize the patterns uncovered in our work at a smaller scale, while still representing viruses from biologically relevant hosts that can contribute to influenza evolution.

Reviewer 2 Report

With the development of technologies such as NGS, a vast amount of virus sequences are being accumulated. Various bioinformatics tools have been developed to analyze them and the comparative analysis of genetic differences is very important for virus evolution and future response. The authors analyzed more than 5000 HA gene sequences of human H1N1 influenza virus using ML, BEAST, metaCATs, HyPhy and Mistic server.
Major comment:
The three 147, 154, and 200 amino acid positions of HA gene are the main of this manuscript. However, it is difficult to be specified through which. The authors stated that their p-value and manual review selected three positions in L217-219 and L314-316. It is not possible to know from the data presented. Why are aa positions with lower p-values (ex. positions 202 and 233 have lower p-value of 10^-20 ~-64) excluded?
1] Is it through the quality score and metrics of ML and BEAST? Table S1 and S2 of supplementary files were not provided to the reviewer and could not be checked.
2] Among the 56 significant amino acid positions of metaCATs, why were only the above three selected?
3] The selection pressures from HyPhy also do not specify the above three amino acid positions.(Table 2).
4] MI value from MISTIC also does not specify the above three amino acid positions.(Table 6)
The rationale for the major comment should be presented and reviewed again.
Minor
1) A high resolution figure of Figure 1 should be provided.
2) Below Table 3, 4 and 5, explanations of abbreviations and meanings in parentheses are required.
3) Numbering is required for supplementary file names.

Author Response

Major Comments

Reviewer Comment 1: The three 147, 154, and 200 amino acid positions of HA gene are the main of this manuscript. However, it is difficult to be specified through which. The authors stated that their p-value and manual review selected three positions in L217-219 and L314-316. It is not possible to know from the data presented. Why are aa positions with lower p-values (ex. positions 202 and 233 have lower p-value of 10^-20 ~-64) excluded? Is it through the quality score and metrics of ML and BEAST? Table S1 and S2 of supplementary files were not provided to the reviewer and could not be checked.

Response: We thank the reviewer for raising these important point. We have numbered the supplementary materials and added tables S1 and S2.                       

The reviewer is correct that we had originally excluded other positions, such as 202 and 233 based on the defined initial criteria. To more objectively identify other potential “drivers” of antigenic drift, we compiled the data from the various statistical analyses and reanalyzed them. We focused on substitutions in annotated functional regions and determined that positions 159 and 233 should be included for their potential role as “drivers'' of antigenic drift. Specifically, position 159 is part of the Lysine fence, while position 233 is in the receptor binding site. We believe that substitutions in such regions could strongly impact influenza evolution and affect its spread in various geographical areas. We have added new text in the Results (sections 3.2 and 3.8) to describe our findings on positions 159 and 233. We have also included a new supplementary table (S16) that summarizes the results from the multiple analyses. We anticipate that this new file will allow readers to better identify other residues that are worth pursuing further.

Position 202 did have a strong p-value, but was located in an alpha helix so it did not meet our criteria. We anticipate that other substitutions, including 202, could be identified in future studies that incorporate wet-lab experiments. We thank the reviewer for prompting us to perform this more in-depth analysis.

Reviewer Comment 2: Among the 56 significant amino acid positions of metaCATs, why were only the above three selected? 

Response: We are grateful that the reviewer asked for more clarity on this matter. Positions 147, 154, and 200 were chosen from the 56 significant meta-CATS results due to their location in functional, structural, and epitope regions.

As mentioned in our response to comment 1 (above), we have re-examined our initial results and concluded that positions 159 and 233 also deserve to be highlighted in this manuscript. We did notice that these two positions displayed some statistical skew in their distribution of residues, which likely artificially lowered the calculated p-value. As such, we still describe them in the results but still primarily focus on positions 147, 154, and 200. We have included all meta-CATS results in a supplementary table (S5) to enable readers to identify other positions that may be worth further experimentation.

Reviewer Comment 3: The selection pressures from HyPhy also do not specify the above three amino acid positions.(Table 2).

Response: We appreciate the reviewer asking for a more in-depth justification for our focus on this subset of residues. We apologize that the text was not clearer on this matter; however, we do report that position 200 is identified in the HyPhy results in the MW and US analyses (section 3.4). We have updated the Discussion section to acknowledge that positions not identified by HyPhy may still play a role in evolution (penultimate paragraph in the Discussion section). Overall, such regions are more conserved--which would prevent them from being identified by this algorithm. We believe that the selection pressures calculated by HyPhy enable us to better differentiate the underlying cause of the mutation--whether due to residues that change frequently, or due to residues that change very rarely.

Reviewer Comment 4: MI value from MISTIC also does not specify the above three amino acid positions.(Table 6). The rationale for the major comment should be presented and reviewed again.

Response: We thank the reviewer for this comment. We apologize for the unclear information in Table 6. We have bolded the four out of five positions in this table that we highlight in our analysis. Briefly, Table 6 does show that positions 147, 154, 200, and 233 are identified by the MISTIC algorithm--even if they generally co-evolve more with other residues rather than with each other. We have added new text to section 3.7 to clarify our findings, including a new observation that positions 147 and 233 were found to strongly co-evolve and were ranked as the forty-first highest pair of residues out of nearly 160,000 pairs in our analysis (supplementary table S15).

Minor Comments

Reviewer Comment 1: A high resolution figure of Figure 1 should be provided.

Response: We agree with the reviewer and have included a higher resolution version of figure 1 to improve readability.

Reviewer Comment 2: Below Table 3, 4 and 5, explanations of abbreviations and meanings in parentheses are required.

Response: We thank the reviewer for bringing this to our attention. We have added explanations of abbreviations and meanings beneath the relevant tables, with an example found below:

“P: Proline; S: Serine; Winter season includes October through March; Summer season includes April through September. MW: Mountain West region: US: USA region; EU: Europe region; NH: Northern Hemisphere region.”

Reviewer Comment 3: Numbering is required for supplementary file names.

Response: We agree with the reviewer and have updated the names of the supplemental files to improve clarity.

Round 2

Reviewer 2 Report

The authors revised the results according to reviewer comments and further analysis.

The selected 5 amino acid postions  significantly changed  in hemagglutinin antigenic sites are presented from the various statistical analyses.

However, a more detailed and further analysis is required. In Table s16, a few positions of Epitope (ex. 62, 177, 277, 299, 319..) seem to show results that meet the author's criteria (despite difficult to know the exact criteria) Why was it excluded?

Table 3-5 and Result(section 3.5) still shows only three amino acid positions.

Is Table 2 necessary for the manuscript page? The data presented by the author do not seem to be helpful in drawing the author's conclusions. It is likely that supplementary data will be sufficient.

Author Response

Comment #1: The authors revised the results according to reviewer comments and further analysis.

Response: We thank the reviewer for their initial insights and comments.

Comment #2: The selected 5 amino acid postions significantly changed in hemagglutinin antigenic sites are presented from the various statistical analyses.

Response: We agree and appreciate the reviewer’s in-depth reading of the initial manuscript.

Comment #3: However, a more detailed and further analysis is required. In Table s16, a few positions of Epitope (ex. 62, 177, 277, 299, 319..) seem to show results that meet the author's criteria (despite difficult to know the exact criteria) Why was it excluded?

      Response: We thank the reviewer for pointing out these additional positions. Our original analysis was focused on regions with functions that were functional and/or structural, but excluded epitopes. We have augmented the text that justifies this decision in the second paragraph of Results subsection 3.3. The new text is below:

We decided to focus on the substitutions at positions 147, 154, and 200 since they are located in functional regions that were not solely characterized as immune epitopes. We consequently excluded the eight positions in known epitopes as well as positions 159 and 233 since we observed a low number of counts for a subset of amino acid residues at these positions adversely skewed statistics and did not justify a more in-depth temporal analysis.

Nevertheless, we agree with the reviewer that additional positions are located in immune epitopes, which could be valuable to include our findings on positions. We have updated the text in the Results (end of subsections 3.2, 3.3, 3,4, 3.5, 3.6, 3.7, and last paragraph in subsection 3.8) as well as the Discussion (second-to-last paragraph) sections. to include positions located in immune epitopes that have significant change over time, significant selection pressure, and high co-evolution scores. These positions include 62, 177, 179, 190, 252, 277, 299, and 313.

Comment #4: Table 3-5 and Result(section 3.5) still shows only three amino acid positions.

      Response: We thank the reviewer for pointing out this perceived inconsistency. We have added text in the second paragraph of Results (subsection 3.3) that justifies our maintained focus on these three positions. Briefly, we expected immune epitope regions to change over time and consequently we had particular interest in regions that had known functional annotations. The new text states:

We decided to focus on the substitutions at positions 147, 154, and 200 since they are located in functional regions that were not immune epitopes. We consequently excluded the eight positions in known epitopes as well as positions 159 and 233 since we observed a low number of counts for a subset of amino acid residues at these positions adversely skewed statistics and did not justify a more in-depth temporal analysis.

We also reiterate this justification in section 3.5. The newly-added justification for excluding other amino acids is below:

We next wanted to determine a more precise evolutionary timeline of the non-epitope positions, or those with skewed p-values, by calculating the frequency of amino acid residues that were at positions 147, 154, and 200.

Comment #5: Is Table 2 necessary for the manuscript page? The data presented by the author do not seem to be helpful in drawing the author's conclusions. It is likely that supplementary data will be sufficient.

      Response: We agree with the reviewer. Since the comprehensive HyPhy results are already available as supplementary material (S6), We have entirely removed Table 2 from the manuscript, and have placed it within a new sheet in the supplementary material S6 file.